# Establishing Joint Orientation Angles of the Limbs in Korean Raccoon Dogs (*Nyctereutes procyonoides koreensis*) Using Computed Tomographic Imaging

**DOI:** 10.3390/ani14192827

**Published:** 2024-09-30

**Authors:** Seongju Ko, Sangjin Ahn, Ho-Hyun Kwak, Heung-Myong Woo, Junhyung Kim

**Affiliations:** 1Department of Veterinary Medicine, Kangwon National University, Chuncheon-si 24341, Gangwon-do, Republic of Korea; tjdwn4936@kangwon.ac.kr (S.K.); kwakhh@kangwon.ac.kr (H.-H.K.); woohm@kangwon.ac.kr (H.-M.W.); 2Gangwon Wildlife Medical Rescue Center, Chuncheon-si 24341, Gangwon-do, Republic of Korea; tkdwls0928@hanmail.net

**Keywords:** *Canidae*, MPR, forelimb, hindlimb

## Abstract

**Simple Summary:**

The Korean raccoon dog (*Nyctereutes procyonoides koreensis*) is an important species in Korea. However, there has been little research about its joint anatomy. In this study, we investigated the anatomy of the limb joints of the Korean raccoon dog using joint orientation angles, a method frequently used in humans, dogs, and cats. The findings of this study should be helpful not only for individual therapies, such as fracture repair, but also for comparative anatomy. Furthermore, this study is the first to measure joint orientation angles in wild animals such as the Korean raccoon dog, making it a useful reference for measurements in other wild canids.

**Abstract:**

Studies are being conducted on the anatomical structures of various wild animals. Despite the ecological importance of the Korean raccoon dog (*Nyctereutes procyonoides koreensis*), limited research has been conducted on its anatomical structure. This study is the first to establish a reference range for joint orientation angles in the limbs of the Korean raccoon dog. Joint orientation angles are an unexplored concept not only in Korean raccoon dogs but also in other wildlife. However, they are important in the examination of the skeletal anatomy of humans and companion animals, such as dogs and cats. Because this type of measurement is still emerging in wildlife research, we applied the methodology used in the domestic dog (*Canis lupus familiaris*). Angles were measured between the mechanical or anatomical axis and the joint orientation lines in the thoracic and pelvic limbs of Korean raccoon dogs. No significant differences were observed between the sexes or between the left and right sides. These findings are consistent with those observed in domestic dogs. Based on this study, a reference range of joint orientation angles could be established for Korean raccoon dogs.

## 1. Introduction

The raccoon dog (*Nyctereutes procyonoides*), a common species in the Far East, originated in the Far East and rapidly spread to Europe [1]. Among the six subspecies of the raccoon dog, the one that inhabits South Korea is *N. p. koreensis.* Despite its modest size, it is becoming the top predator in Korea because of a decline in other predators and competitors [2,3]. For these reasons, the Korean raccoon dog (*Nyctereutes procyonoides koreensis*) is ecologically important in Korea. The Korean raccoon dog is a distinct population that has adapted to the particular habitat of the Korean Peninsula, and it differs from other continental and Japanese raccoon dog populations, according to phylogeographic research using complete mtDNA cytochrome b sequences [4].

Previous studies on Korean raccoon dogs have largely concentrated on genetic research and the diseases they transmit [3,4,5,6]. Although these studies have improved our understanding of the role of Korean raccoon dogs in the ecosystem, a significant knowledge gap exists regarding their anatomy. Imaging techniques have been used to determine the normal skeletal morphology of the limbs in other wild species, such as anteaters, lions, and meerkats [7,8,9]. However, while studies have been conducted on the dissection of the aortic arch, forearms, and skulls of Korean raccoon dogs, no research has been performed to define skeletal reference ranges for their limbs [10,11,12].

This study aimed to establish an anatomical reference range for the limbs of Korean raccoon dogs using joint orientation angles. In this technique, the joint orientation lines and mechanical or anatomical axes of the bone are combined to generate joint orientation angles. In humans and companion animals, these angles are important for the repair of fractures and the correction of deformities. In other words, the joint orientation angles are used to assess limb alignment. Abnormalities in the limb alignment indicate deformities. Such deformities can lead to not only cosmetic but also functional abnormalities.

Because joint orientation angles have not been measured in Korean raccoon dogs or other wild animals, no research has been conducted on the diagnosis and treatment of their deformities. Thus, the aim of this study was to establish reference ranges for joint orientation angles in Korean raccoon dogs. Studies measuring joint orientation angles in domestic dogs were consulted to define the computed tomography (CT) technique and the values to be measured [13,14,15,16,17,18,19,20,21,22].

## 2. Materials and Methods

### 2.1. Animals

CT data on the thoracic (*n* = 20) and pelvic limbs (*n* = 20) of 10 raccoon dogs (six males and four females) that were rescued and brought to the Gangwon Wildlife Medical Rescue Center at Kangwon National University from 2022 to 2023 were collected.

This study was approved by the Institutional Animal Care and Use Committee of Kangwon National University, Chuncheon-si, Gangwon-do, Republic of Korea (Approval No. KW-240109-3).

### 2.2. CT Technique

All CT images were obtained using a 16-slice helical CT scanner with a 1 mm slice thickness (Alexion, Canon, Japan). CT imaging was conducted under general anesthesia. Isoflurane (Ifran soln, Hana Pharm, Korea) was used to maintain anesthesia, and the animals were positioned in ventral recumbency for imaging. Three-dimensional (3D) reconstruction was subsequently performed using the software (3D slicer, www.slicer.org). Multiplanar reconstruction (MPR) was also performed using the software. To aid in the identification of the bone margins, a maximum-intensity projection (MIP) was created for some images by increasing the slice thickness. Based on the anatomical landmarks, the joint orientation angles of the thoracic and pelvic limbs were measured in the frontal, sagittal, and axial planes. All measurements were performed by a single veterinarian.

### 2.3. Measurements

#### 2.3.1. Mechanical Medial Proximal Humeral Angle (mMPHA) and Mechanical Lateral Distal Humeral Angle (mLDHA)

First, the frontal plane was set using MPR. The MIP was then used to help set up the frontal plane. The midpoint of the oval that fits the proximal humeral head and the distal articular surface of the humeral condyle were connected to form the mechanical axis in the MIP of the frontal plane [13]. In addition, this mechanical axis was matched with the orientation line of the frontal plane image’s sagittal plane. In this aligned state, the frontal plane images were scrolled to find the planes containing the proximal and distal joint orientation lines of the humerus. The mechanical medial proximal humeral angle (mMPHA) and mechanical lateral distal humeral angle (mLDHA) were determined through measurement of the angles produced by the mechanical axis, proximal joint orientation line, and distal joint orientation line of the humerus (Figure 1A,B) [14].

#### 2.3.2. Mechanical Caudal Proximal Humeral Angle (mCdPHA) and Mechanical Cranial Distal Humeral Angle (mCrDHA)

By connecting the center of the oval that precisely fits the humeral head proximally to the center of the circle that precisely fits the lateral humeral condyle distally, the mechanical axis in the MIP of the sagittal plane could be drawn [13]. This mechanical axis was aligned with the frontal plane orientation line in the sagittal plane image. In this state, the sagittal plane images were scrolled to identify the planes containing the proximal and distal joint orientation lines of the humerus. The mechanical caudal proximal humeral angle (mCaPHA), which is the angle between the mechanical axis and the proximal joint orientation line of the humerus, and the mechanical cranial distal humeral angle (mCrDHA), which is the angle between the mechanical axis and the distal joint orientation line of the humerus, were measured (Figure 1C,D) [14].

#### 2.3.3. Humeral Torsional Angle (HTA)

In the axial plane, the proximal humeral axis (PHA) was drawn by connecting the most caudal eminence of the greater tubercle (CdGT) and the tubercle located at the center of the caudomedial line of the humeral head (HHT). The distal humeral trochlear axis (DHTA) was then drawn by connecting the most caudal points of the medial and lateral ridges of the trochlea. The humeral torsional angle (HTA), which is the angle formed by the PHA and DHTA, was measured. The external and internal torsions were indicated by positive and negative values, respectively (Figure 1E) [14].

#### 2.3.4. Anatomical Medial Proximal Radial Angle (aMPRA) and Anatomical Lateral Distal Radial Angle (aLDRA)

The frontal plane was set using MPR. The anatomical axis was drawn by connecting the midpoints of the diaphysis at 25, 50, and 75% of the radius in the MIP of the frontal plane [23]. Then, the sagittal plane orientation line was aligned with this anatomical axis. After alignment, the frontal plane images were scrolled thoroughly to position them on the planes, including the proximal and distal joint orientation lines of the radius. The anatomical medial proximal radial angle (aMPRA) and the anatomical lateral distal radial angle (aLDRA) were the angles created by the anatomic axis and the proximal and distal joint orientation lines of the radius, respectively (Figure 2A,B) [14].

#### 2.3.5. Anatomical Cranial Proximal Radial Angle (aCrPRA) and Anatomical Caudal Distal Radial Angle (aCdDRA)

In the sagittal plane, the anatomical axis is curved owing to the natural cranial bowing of the canine radius. Therefore, the anatomic axis was divided into proximal and distal parts. In the MIP of the sagittal plane, the proximal anatomical axis was drawn by connecting the midpoints of the diaphysis at 25 and 50% of the proximal radius [15]. Then, the frontal plane orientation line was aligned with this proximal anatomic axis. In this state, the sagittal plane images were scrolled through to find the plane containing the proximal joint orientation line of the radius. We then measured the angle formed by the proximal anatomic axis and the proximal joint orientation line of the radius, which is the anatomical cranial proximal radial angle (aCrPRA). Similarly, we drew the distal anatomic axis by connecting the midpoints of the diaphysis at 25% and 50% of the distal radius, and we measured the angle formed by the distal anatomic axis and the distal joint orientation line of the radius, which is the anatomical caudal distal radial angle (aCdDRA) (Figure 2C,D) [14,15].

#### 2.3.6. Radial Torsional Angle (RTA)

In the axial plane, a tangential line was drawn at the cranial border of the radial head fovea. A line connecting the cranial eminence of the ulnar notch and the groove for the long abductor muscle of the first digit was drawn. The angle formed between these two lines was the radial torsional angle (RTA). The external and internal torsions were indicated by positive and negative values, respectively (Figure 2E) [16].

#### 2.3.7. Anatomical Lateral Proximal Femoral Angle (aLPFA) and Anatomical Lateral Distal Femoral Angle (aLDFA)

The femoral MPR was aligned as described by Barnes et al. [17]. After producing an MIP, the anatomical axis was drawn by connecting the midpoints of the diaphysis at one-third and one-half of the femur. Frontal plane images were then scanned to find the center of the femoral head and the tip of the greater trochanter. A hip-joint orientation line was drawn by connecting the two points. The anatomical lateral proximal femoral angle (aLPFA), which is the angle created by the hip-joint orientation line and the anatomical axis, was measured. To measure the anatomical lateral distal femoral angle (aLDFA), a knee-joint orientation line was drawn connecting the most distal points of the medial and lateral femoral condyles. The angle between the knee-joint orientation line and the anatomical axis was measured (Figure 3A) [17,18,24].

#### 2.3.8. Mechanical Lateral Proximal Femoral Angle (mLPFA) and Mechanical Lateral Distal Femoral Angle (mLDFA)

The frontal plane images were scanned to identify the center of the femoral head and the femoral intercondylar notch. The mechanical axis was drawn by connecting the two points. The mechanical lateral proximal femoral angle (mLPFA), which is the angle between the mechanical axis and the hip-joint orientation line, and the mechanical lateral distal femoral angle (mLDFA), which is the angle between the mechanical axis and the knee-joint orientation line, were measured (Figure 3B) [18,24].

#### 2.3.9. Neck Shaft Angle

In the frontal plane, a line was drawn connecting the proximal epiphysis of the femur to the midpoint of the femur head, following the method described by Lusetti et al. [18]. The angle formed between this line and the anatomical axis, known as the neck shaft angle (NSA), was measured (Figure 3C) [18,24].

#### 2.3.10. Anatomical Caudal Proximal Femoral Angle (aCdPFA), Anatomical Caudal Distal Femoral Angle (aCdDFA), and Procurvation Angle (PA)

In the sagittal plane, the anatomical axis of the femur is curved. Therefore, the anatomical axis was divided into proximal and distal parts. The anatomical caudal proximal femoral angle (aCdPFA) was measured by finding the plane containing the proximal joint orientation line and measuring the angle formed by the proximal anatomic axis (PAA) and the mid-diaphyseal line of the neck of the femur. Sagittal plane images were scrolled to determine the distal anatomical axis (DAA), and the anatomical caudal distal femoral angle (aCdDFA) was measured. The lesser trochanter and trochlear limit were then connected, and the perpendicular angle generated by the line connecting these two sites was measured. The procurvation angle (PA) formed by the PAA and DAA was measured (Figure 3D–F) [19,24].

#### 2.3.11. Anteversion Angle (AA)

In the axial plane, a line was drawn across the center of the femoral head and the center of the femoral neck. Next, a line was drawn tangential to the medial and lateral femoral condyles. The anteversion angle was the angle generated by these two lines (AA) (Figure 3G) [18,20].

#### 2.3.12. Mechanical Medial Proximal Tibial Angle (mMPTA) and Mechanical Medial Distal Tibial Angle (mMDTA)

The MPR of the tibia was aligned as described by Barnes et al. The images were scrolled to find the most distal part of the intermediate ridge of the tibial cochlea as well as the midpoint of the medial and lateral intercondyloid eminences. The mechanical axis was drawn by connecting these two points. The mechanical medial proximal tibial angle (mMPTA), which is the angle created by the knee-joint orientation line and the mechanical axis, was measured. The mechanical medial distal tibial angle (mMDTA), which is the angle is created by the ankle joint orientation line and mechanical axis, was also measured. The knee-joint orientation line was formed by connecting the two concave points of the tibial plateau subchondral line. The ankle joint orientation line was formed by connecting the distal points of the medial and lateral condyle concavities (Figure 4A) [17,18,24].

#### 2.3.13. Mechanical Caudal Proximal Tibial Angle (mCdPTA), Mechanical Cranial Distal Tibial Angle (mCrDTA), and Tibial Plateau Angle (TPA)

The mechanical axis in the sagittal plane was drawn by connecting the trochlear center of the talus with the center of the intercondylar eminence. A line connecting the cranial and caudal margins of the tibial plateau was drawn. The mechanical caudal proximal tibial angle (mCdPTA) was the angle formed between this line and the mechanical axis. A line was drawn connecting the most distal parts of the cranial and caudal tibia, and the angle formed by this line and the mechanical axis, which is the mechanical caudal distal tibial angle (mCdDTA), was measured [18]. To find the tibial plateau angle (TPA), the mCdDTA was subtracted from 90° (Figure 4B).

#### 2.3.14. Tibial Torsional Angle (TTA)

In the axial plane, the tibial torsional angle (TTA) was measured as the angle between the transcondylar (TC) and distal cranial tibial (CnT) axes. The TC axis runs from the prominence of the medial collateral ligament insertion to the caudolateral extension of the extensor sulcus. The distal CnT axis runs parallel to the cranial cortex of the tibia and is directly proximal to the talocrural joint (Figure 4C) [21,22].

### 2.4. Statistical Analysis

All statistical analyses were performed using SPSS (version 29.0.2.0; IBM Corporation, Armonk, NY, USA). Data are presented as the mean ± standard deviation (SD). Differences between sexes or the left and right sides were examined. The Shapiro–Wilk test was used to determine whether the data were normal. When normality was not fulfilled, Mann–Whitney U tests were used to compare the joint orientation angle data by sex or by left and right side. Otherwise, independent t-tests were used. Statistical significance was set at *p* < 0.05.

## 3. Results

Of the 10 raccoon dogs, 6 were male and 4 were female. All measurements were compared between sexes, but no significant differences were observed. Measurements of the right (*n* = 10) and left thoracic limbs (*n* = 10) were compared; however, no significant differences were observed. The measurements of the right (*n* = 10) and left pelvic limbs (*n* = 10) were also compared, and no significant differences were observed. The means ± standard deviation (SDs) of the joint orientation angles of the thoracic and pelvic limbs of the raccoon dogs are shown in Table 1 and Table 2, respectively.

## 4. Discussion

This study aimed to establish normal skeletal reference ranges for the limbs of Korean raccoon dogs. No previous research has been conducted using CT to determine normal skeletal reference ranges for the limbs of any wildlife, including Korean raccoon dogs. This has led to difficulties in the diagnosis and treatment of angular deformities in wild populations. Therefore, in this study, we established anatomical reference ranges for the limbs of Korean raccoon dogs. As these have never been measured in Korean raccoon dogs, methods used for other animals were applied [13,14,15,16,17,18,19,20,21,22]. Dogs and raccoon dogs both belong to the *Canidae* family and are genetically similar; because many studies measuring joint orientation angles have been performed in domestic dogs, we applied similar methods to measure the joint orientation angles in raccoon dogs. In this study, we found similarities in the joint orientation angles between the two species [25,26].

We used the MPR-CT method to measure the joint orientation angles in this study. Two main methods are used to measure joint orientation angles: radiography and MPR-CT. MPR-CT allows for more accurate measurements than radiography, because the MPR can manually adjust the plane, allowing for better identification of anatomical points and more accurate setting of the anatomical and mechanical axes. Although sedation and CT equipment are required to obtain CT images, MPR-CT was selected in this study to establish a more accurate reference range.

We used measurement techniques from studies on dogs for both the thoracic and pelvic limbs. However, unlike the pelvic limbs, only one previous study had used MPR to measure joint orientation angles in the thoracic limbs. Therefore, for the thoracic limb measurements, the measurement method used in that single study was used to measure the joint orientation angles of the shoulder, elbow, and carpus [14]. For the pelvic limbs, we compared several studies and adopted one of the methods. We primarily used the method described in a previous study that researched repeatability with minimal inter-measurement error [17].

The measured values were compared with those from studies that measured the joint orientation angles in domestic dogs (Table 3 and Table 4). Most of the values obtained for Korean raccoon dogs and domestic dogs were similar, but a few differences were noted. The mCrDHA was larger in Korean raccoon dogs than in domestic dogs. This demonstrates that the humeral epicondyles of Korean raccoon dogs differ from those of domestic dogs [13,14]. The RTA was larger in Shih Tzu dogs than in Korean raccoon dogs. This may be because the Shih Tzu breed has more deformities than other canine breeds [14,16]. The TTA was smaller in this Korean raccoon dog study than in two other studies on domestic dogs. This indicates that torsion of the tibia is almost absent in Korean raccoon dogs [18,19]. This anatomical study helped to identify similarities and differences in the extremities of different *Canidae* species, and further studies on other breeds could provide insights into congenital orthopedic conditions, such as angular deformities and medial patellar luxation, in dogs.

This study has several limitations. First, the measurements were conducted only on individuals rescued by the Gangwon Wildlife Rescue Center. A genetic study on Korean raccoon dogs for disease management classified them into four management units: northern, central, southwestern, and southeastern regions [3]. The Korean raccoon dogs used in this study were assigned to the northern management unit, suggesting that the sample size may not be representative of the total population. Second, the methods for obtaining joint orientation lines, anatomical axes, and mechanical axes were derived from domestic dog studies. Some values obtained using these methods may be irrelevant to Korean raccoon dogs, and other important values may be yet to be discovered for raccoon dogs. Third, the measurements were conducted by a single measurer, which could have introduced bias.

This work is important because it is the first to measure the anatomical reference range of limbs in Korean raccoon dogs, despite these restrictions. Joint orientation angles are important indicators, especially for fracture correction. When fractures occur in both limbs, reference ranges are required to ensure functional recovery through accurate fracture repair. Functional recovery is even more critical for wildlife than for companion animals. If functional recovery fails, these animals may be forced to spend their entire lives in human care centers instead of returning to their natural habitats. Furthermore, for predators such as the Korean raccoon dog, poor limb function can seriously impair their hunting ability, which is critical for their survival.

Therefore, there is an increasing need for research on the anatomy of wildlife. First, an increasing number of people are adopting wildlife as companion animals [27]. Moreover, the number of collisions between motor vehicles and wildlife is increasing [28,29]. An understanding of the anatomy of these animals must be acquired for their effective treatment. From a veterinary perspective, knowledge of the anatomical structures of wildlife aids in the diagnosis of wounds, illnesses, and other health problems, thereby facilitating the treatment of particular animals. Furthermore, through comparative anatomy, anatomical studies of animals improve our knowledge of not only one species but also other species. Evolutionary biology is aided by comparative anatomy, which reveals phylogenetic links between species [30,31]. Wildlife anatomical research must continue for these goals to be achieved. This study improved our understanding of the anatomy of Korean raccoon dogs. We expect to conduct further research on the joint orientation angles in other wildlife species.

## 5. Conclusions

This study provides baseline data on the joint orientation angles of Korean raccoon dogs for use in veterinary medicine. To the best of our knowledge, no other studies have established a reference range for limb joint orientation angles in Korean raccoon dogs. We anticipate that this will facilitate further studies on wildlife anatomy.

## Figures and Tables

**Figure 1 animals-14-02827-f001:**
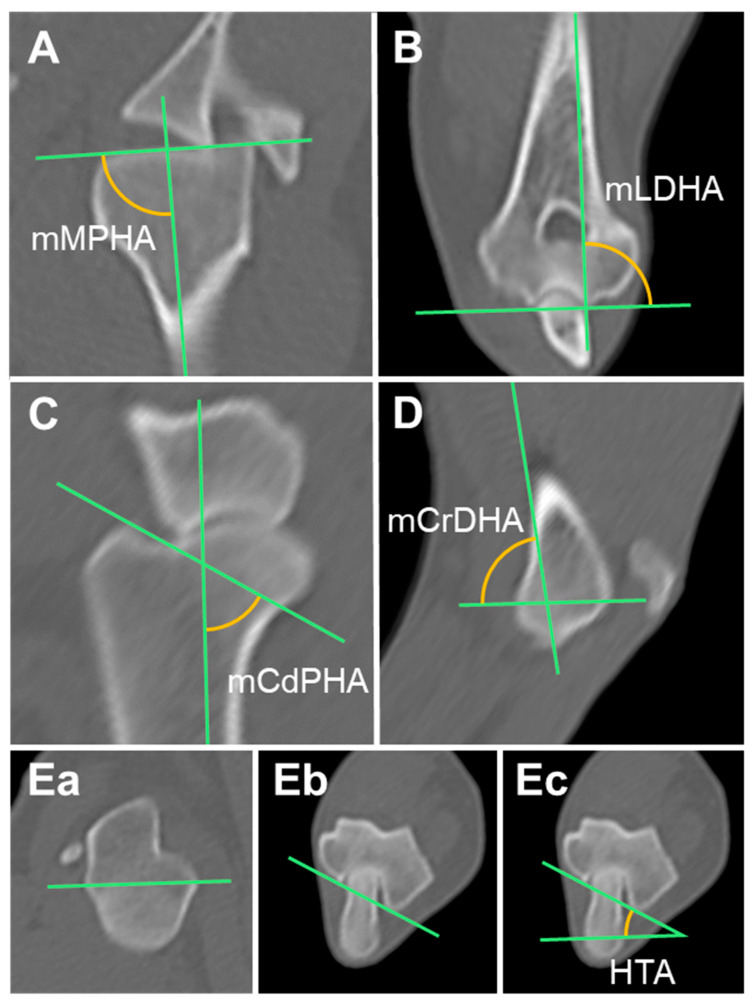
Measurement of the humerus joint orientation angles in multiplanar reconstruction computed tomography (MPR CT): (**A**,**B**) Mechanical axis and the humeral angles in the frontal plane. (**C**,**D**) Mechanical axis and the humeral angles in the sagittal plane. (**Ea**,**Eb**,**Ec**) Humeral angle in the axial plane. mMPHA, mechanical medial proximal humeral angle; mLDHA, mechanical lateral distal humeral angle; mCdPHA, mechanical caudal proximal humeral angle; mCrDHA, mechanical cranial distal humeral angle; HTA, humeral torsional angle.

**Figure 2 animals-14-02827-f002:**
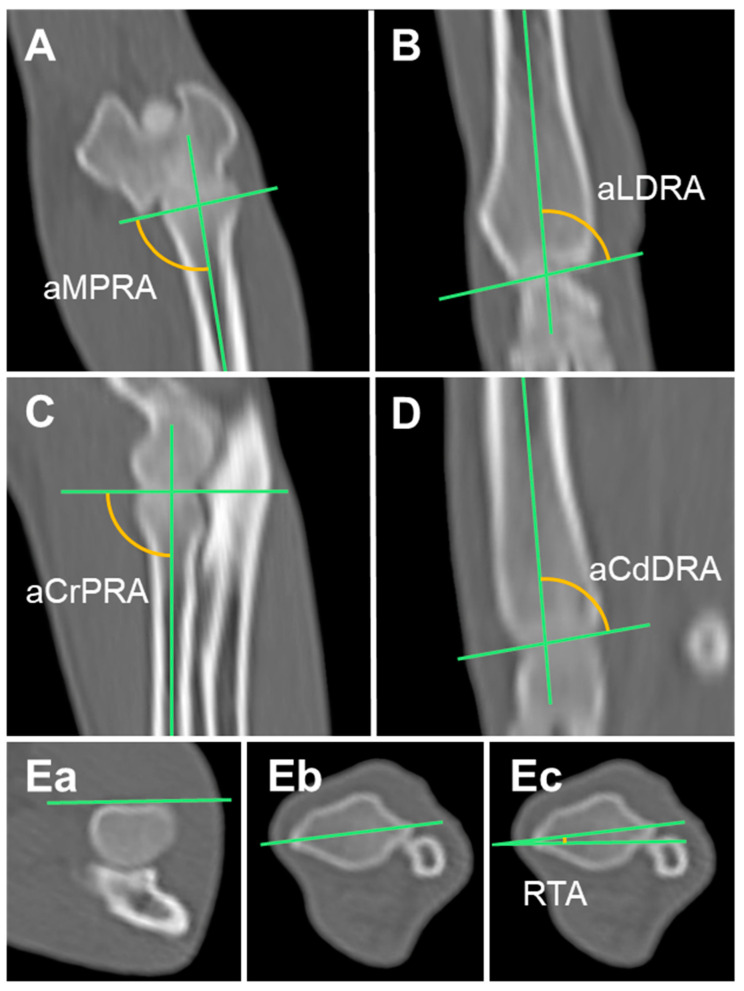
Measurement of the radial joint orientation angles in multiplanar reconstruction computed tomography (MPR CT): (**A**,**B**) Anatomical axis and the radial angles in the frontal plane. (**C**,**D**) Anatomical axis and the radial angles in the sagittal plane. (**Ea**,**Eb**,**Ec**) Radial angle in the axial plane. aMPRA, anatomical medial proximal radial angle; aLDRA, anatomical lateral distal radial angle; aCrPRA, anatomical cranial proximal radial angle; aCdDRA, anatomical caudal radial angle; RTA, radial torsional angle.

**Figure 3 animals-14-02827-f003:**
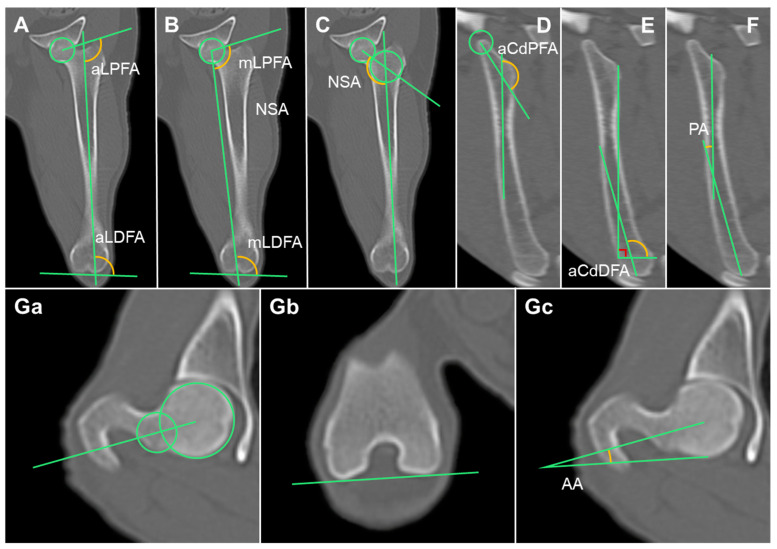
Measurement of the femur joint orientation angles in multiplanar reconstruction computed tomography (MPR CT): (**A**–**C**) Anatomical and mechanical axes and the femoral angles in the frontal plane. (**D**–**F**) Anatomical axis and the femoral angles in the sagittal plane. (**Ga**,**Gb**,**Gc**) Femoral angle in the axial plane. aLPFA, anatomical lateral proximal femoral angle; aLDFA, anatomical distal femoral angle; mLPFA, mechanical lateral proximal femoral angle; mLDFA, mechanical lateral distal femoral angle; NSA, neck shaft angle; aCdPFA, anatomical caudal proximal femoral angle; aCdDFA, anatomical caudal distal femoral angle; PA, procurvation angle; AA, anteversion angle.

**Figure 4 animals-14-02827-f004:**
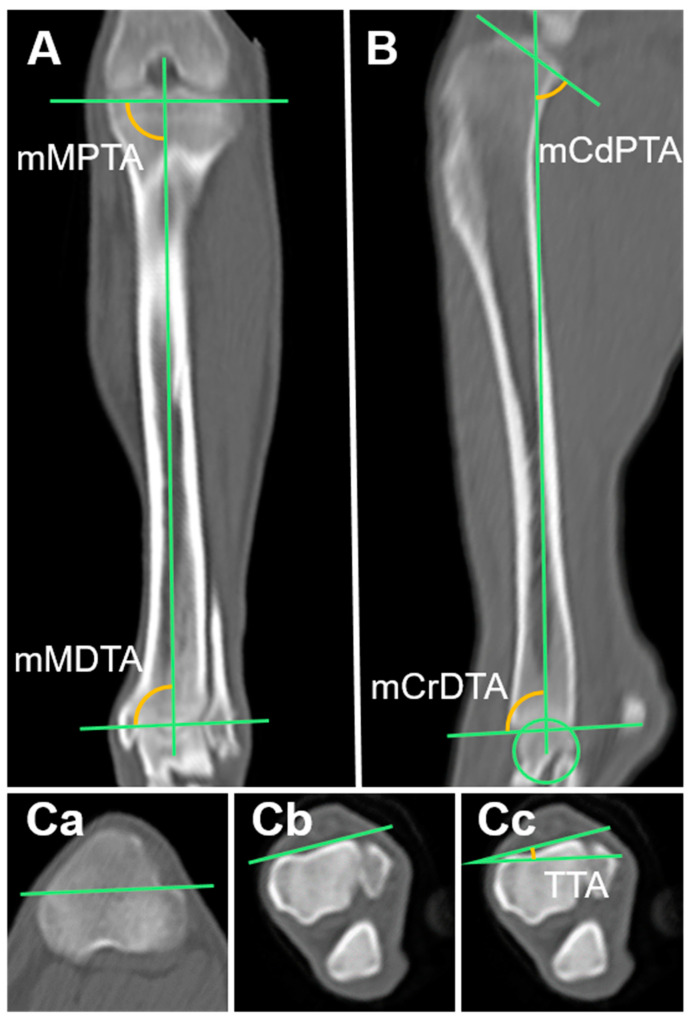
Measurement of the tibia joint orientation angles in multiplanar reconstruction computed tomography (MPR CT): (**A**) Mechanical axis and the tibial angles in the frontal plane. (**B**) Anatomical axis and the tibial angles in the sagittal plane. (**Ca**,**Cb**,**Cc**) Tibial angle in the axial plane. mMPTA, mechanical medial proximal tibial angle; mMDTA, mechanical medial distal tibial angle; mCdPTA, mechanical caudal proximal tibial angle; mCrDTA; mechanical cranial distal tibial angle; TTA, tibial torsional angle.

**Table 1 animals-14-02827-t001:** The thoracic limb angles calculated using CT measurements in the Korean raccoon dogs.

Humeral Joint Orientation Angles (°)	Measurements			
	Mean	SD	Min	Max
mMPHA (°)	90.49	3.65	93.00	97.70
mLDHA (°)	87.58	4.67	80.90	101.00
mCdPHA (°)	54.23	7.87	37.10	66.70
mCrDHA (°)	104.12	10.98	81.00	123.60
HTA (°)	22.24	5.55	13.70	34.30
**Radial** **Joint Orientation Angles (°)**	**Measurements**			
	**Mean**	**SD**	**Min**	**Max**
aMPRA (°)	88.12	5.99	74.60	100.60
aLDRA (°)	89.70	5.15	77.00	98.40
aCrPRA (°)	87.30	4.72	72.90	92.50
aCdDRA (°)	85.36	4.59	75.40	95.40
RTA (°)	1.17	6.81	−11.50	8.50

mMPHA, mechanical medial proximal humeral angle; mLDHA, mechanical lateral distal humeral angle; mCdPHA, mechanical caudal proximal humeral angle; mCrDHA, mechanical cranial distal humeral angle; HTA, humeral torsional angle; aMPRA, anatomical medial proximal radial angle; aLDRA, anatomical lateral distal radial angle; aCrPRA, anatomical cranial proximal radial angle; aCdDRA, anatomical caudal radial angle; RTA, radial torsional angle.

**Table 2 animals-14-02827-t002:** The pelvic limb angles calculated using CT measurements in the Korean raccoon dogs.

Femoral Joint Orientation Angles (°)	Measurements			
	Mean	SD	Min	Max
aLPFA (°)	108.37	5.81	97.40	123.30
aLDFA (°)	92.94	4.02	85.40	101.80
NSA (°)	132.16	8.13	117.40	144.20
mLPFA (°)	103.43	5.76	91.30	117.10
mLDFA (°)	97.93	3.78	90.00	107.00
aCdPFA (°)	141.18	13.72	109.30	166.10
aCdDFA (°)	104.46	2.80	101.00	109.50
PA (°)	9.26	3.99	3.90	20.70
AA (°)	25.15	6.03	14.80	35.20
**Tibial** **Joint Orientation Angles (°)**	**Measurements**			
	**Mean**	**SD**	**Min**	**Max**
mMPTA (°)	91.40	2.77	87.10	96.00
mMDTA (°)	91.43	3.27	87.30	98.90
mCdPTA (°)	57.76	5.74	47.50	71.50
mCrDTA (°)	90.51	4.10	82.10	96.90
TPA (°)	31.75	4.88	23.70	42.50
TTA (°)	0.49	5.79	−11.40	8.40

aLPFA, anatomical lateral proximal femoral angle; aLDFA, anatomical distal femoral angle; mLPFA, mechanical lateral proximal femoral angle; mLDFA, mechanical lateral distal femoral angle; NSA, neck shaft angle; aCdPFA, anatomical caudal proximal femoral angle; aCdDFA, anatomical caudal distal femoral angle; PA, procurvation angle; AA, anteversion angle; mMPTA, mechanical medial proximal tibial angle; mMDTA, mechanical medial distal tibial angle; mCdPTA, mechanical caudal proximal tibial angle; mCrDTA; mechanical cranial distal tibial angle; TTA, tibial torsional angle.

**Table 3 animals-14-02827-t003:** Comparison of the thoracic limb angles of canines.

	Raccoon Dogs	Shih Tzus [14]	Canine Cadavers [13]	Poodles [23]	10 Normal Dogs [16]
	CT	CT	Radiograph	Radiograph	CT
**Humerus (°)**					
mMPHA (°)	90.49 ± 3.65	84.74 ± 3.95	-	-	-
mLDHA (°)	87.58 ± 4.67	85.04 ± 2.57	86.92 ± 1.24	-	-
mCdPHA (°)	54.23 ± 7.87	46.75 ± 2.20	43.28 ± 5.44	-	-
mCrDHA (°)	104.12 ± 10.98	79.47 ± 1.97	71.86 ± 3.97	-	-
HTA (°)	22.24 ± 5.55	19.16 ± 2.38	-	-	27.4
**Radius (°)**					
aMPRA (°)	88.12 ± 5.99	85.04 ± 1.58	-	77.91 ± 3.44	-
aLDRA (°)	89.70 ± 5.15	87.59 ± 1.37	-	89.60 ± 2.03	-
aCrPRA (°)	87.30 ± 4.72	84.60 ± 1.16	-	88.13 ± 3.91	-
aCdDRA (°)	85.36 ± 4.59	84.27 ± 1.79	-	71.11 ± 3.80	-
RTA (°)	1.17 ± 6.81	20.91 ± 3.00	-	-	2.7

mMPHA, mechanical medial proximal humeral angle; mLDHA, mechanical lateral distal humeral angle; mCdPHA, mechanical caudal proximal humeral angle; mCrDHA, mechanical cranial distal humeral angle; HTA, humeral torsional angle; aMPRA, anatomical medial proximal radial angle; aLDRA, anatomical lateral distal radial angle; aCrPRA, anatomical cranial proximal radial angle; aCdDRA, anatomical caudal radial angle; RTA, radial torsional angle.

**Table 4 animals-14-02827-t004:** Comparison of the pelvic limb angles of canines.

	Raccoon Dogs	English Bulldogs [18]		Poodles [19]
	CT	CT	Radiograph	CT
**Femur (°)**				
aLPFA (°)	108.37 ± 5.81	111.75 ± 6.66	106.6 ± 8.7	119.5 ± 5.7
aLDFA (°)	92.94 ± 4.02	92.33 ± 4.75	94.4 ± 4.1	90.3 ± 2.8
NSA (°)	132.16 ± 8.13	129.11 ± 8.03	127.7 ± 6.3	116.8 ± 6.1
mLPFA (°)	103.43 ± 5.76	111.02 ± 6.90	102.1 ± 8.8	113.6 ± 6.1
mLDFA (°)	97.93 ± 3.78	101.56 ± 2.43	99.1 ± 3.1	96.2 ± 2.5
aCdPFA (°)	141.18 ± 13.72	-	157.3 ± 7.7	153.3 ± 5.1
aCdDFA (°)	104.46 ± 2.80	-	104.3 ± 2.1	102.9 ± 3.2
PA (°)	9.26 ± 3.99	-	12.7 ± 4.1	11.2 ± 5.2
AA (°)	25.15 ± 6.03	-	-	20.8 ± 4.1
**Tibia (°)**				
mMPTA (°)	91.40 ± 2.77	91.98 ± 4.28	94.4 ± 3.8	92.8 ± 2.1
mMDTA (°)	91.43 ± 3.27	91.34 ± 2.98	96.5 ± 2.3	96.5 ± 4.1
mCdPTA (°)	57.76 ± 5.74	63.25 ± 6.15	-	-
mCrDTA (°)	90.51 ± 4.10	86.73 ± 3.52	91.0 ± 4.6	98.5 ± 3.8
TPA (°)	31.75 ± 4.88	-	27.6 ± 4.7	21.3 ± 3.3
TTA (°)	0.49 ± 5.79	4.00 ± 8.82	-	11.3 ± 4.3

aLPFA, anatomical lateral proximal femoral angle; aLDFA, anatomical distal femoral angle; mLPFA, mechanical lateral proximal femoral angle; mLDFA, mechanical lateral distal femoral angle; NSA, neck shaft angle; aCdPFA, anatomical caudal proximal femoral angle; aCdDFA, anatomical caudal distal femoral angle; PA, procurvation angle; AA, anteversion angle; mMPTA, mechanical medial proximal tibial angle; mMDTA, mechanical medial distal tibial angle; mCdPTA, mehcanical caudal proximal tibial angle; mCrDTA; mechanical cranial distal tibial angle; TTA, tibial torsional angle.

## Data Availability

All of the data are discussed in the manuscript.

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
