# Peer review of "Establishing Joint Orientation Angles of the Limbs in Korean Raccoon Dogs (Nyctereutes procyonoides koreensis) Using Computed Tomographic Imaging"

_animals, 2024, doi:10.3390/ani14192827_

Round 1
Reviewer 1 Report
Comments and Suggestions for Authors
The article entitled “Establishing joint orientation angles in Korean raccoon dogs (Nyctereutes procyonoides koreensis) using computed tomographic imaging” is an applied anatomical study of the main joints of the thoracic and pelvic limbs, such as shoulder, elbow, hip, and knee. Interestingly, it compares with other studies on the domestic dog (Canis lupus familiaris) in the discussion section since it is the species base of veterinary procedures in wildlife medicine. For this manuscript, I only have some minor revisions. The veterinary justification is sufficient and it is not necessary to cite studies in humans because Homo sapiens is a species phylogenetically distant of the canids. Besides, it is a bipedal animal where the angles of the limb joints are very different. Please specify the bones and the sides in the figures (right or left shoulder joint, right or left elbow joint). It would be useful for the readers to include the names of the bone parts (e.g. lateral and medial condyles of the femur, humeral trochlea, radial head, trochlear incisure, etc) in each subfigure. Change the term “extremity” to “limb” or its respective plural form along the manuscript. Change the terms of the planes “frontal” and “axial” to “dorsal” and “transverse”, respectively along the manuscript.
The minor revisions are line by line as below:
1- Please specify the studied joints in the title. E.g. Establishing joint orientation angles of the limbs in…
12- However, there has been little research about their joint anatomy.
13. change “the anatomical structure of the extremities” to “the anatomy of the limb joints..”
16, 60, 61, 280, 281: Are you sure that your study is the first in wild animals or in Korean Raccoon dog? Are you taking into account the articles around the world? Or it is from the knowledge of the authors?
17-18: in other wild canids. The wild species are very variant.
26- …used in the domestic dog…
32- Please replace the words to others that are not in the title. For example: Canidae, elbow, hip, knee, shoulder.
81- change the terms of the planes “frontal” and “axial” to “dorsal” and “transverse”, respectively along the manuscript.
Figure 1. It would be useful for the readers to know the names of the bones in each subfigure. Please specify the side of the figure (right or left shoulder joint, right or left elbow joint)
Figure 2.
157. Is the cranial eminence the medial coronoid process of the ulna? and the medial eminence is the lateral coronoid process? If they are those structures please use the anatomically correct names.
177- Incisure femoral or intertrochanteric incisure?
285-288. Change the redaction of this paragraph. Both species are close phylogenetically because are of the Family Canidae and your study found a similar joint limb angulation in the anatomy between both species Canis lupus familiaris and Nyctereutes procyonoides.
Reviewer 2 Report
Comments and Suggestions for Authors
Summary: The aim of this paper was to establish an anatomical reference range for the shoulder, elbow, hip, and stifle of 53 Korean raccoon dogs using joint orientation angles obtained vis CT scan. This paper provides the first accounting of the average joint angles for the Korean racoon dog which can aid in the correction of angular limb deformities and fracture fixation.
Line 82 – Which author? was vet performing the measurements a boarded radiologist? What qualifications do they have to make these measurements
Line 86 – correct grammar “First, the frontal plane was set”
Figure 1 A and B: would prefer a more zoomed out image showing the entire anatomic axis connecting the center of the humeral head with the humeral condyles to justify the presented mechanical axis. Also, would prefer to see an image with better definition of for humeral condyles not just alignment of the trochlea
Figure 1 D: insufficient view to demonstrate sagittal angle
Figure 4 B: add diagram of mCdDTA and TPA to figure
Line 280: correct to say “No previous research has been conducted using CT to determine..”
Line 300: measurements were performed on the distal radius but not specifically the carpal joints
Line 317: Correct table title to mention values compared with mean and SD values
Other notes: Add description of sedation or anesthetic protocol used and animal positioning during CT scanning
Comments on the Quality of English LanguageA few minor corrections for more concise wording were found. Overall very good English
